# Exploiting the Nucleophilicity of the Nitrogen Atom of Imidazoles: One-Pot Three-Component Synthesis of Imidazo-Pyrazines

**DOI:** 10.3390/molecules24101959

**Published:** 2019-05-21

**Authors:** Ubaldina Galli, Rejdia Hysenlika, Fiorella Meneghetti, Erika Del Grosso, Sveva Pelliccia, Ettore Novellino, Mariateresa Giustiniano, Gian Cesare Tron

**Affiliations:** 1Dipartimento di Scienze del Farmaco, Università del Piemonte Orientale “A. Avogadro”, 28100 Novara, Italy; ubaldina.galli@uniupo.it (U.G.); 10026282@studenti.uniupo.it (R.H.); erika.delgrosso@uniupo.it (E.D.G.); 2Dipartimento di Scienze Farmaceutiche, Università degli Studi di Milano, 20133 Milano, Italy; fiorella.meneghetti@unimi.it; 3Dipartimento di Farmacia, Università degli Studi di Napoli “Federico II”, 80131 Napoli, Italy; sveva.pelliccia@unina.it (S.P.); ettore.novellino@unina.it (E.N.)

**Keywords:** multicomponent reactions, interrupted Ugi reactions, isocyanides

## Abstract

A novel one-pot multicomponent reaction to synthesize substituted imidazopyrazines is described. In brief, 1*H*-(imidazol-5-yl)-*N*-substituted methanamines react with aldehydes and isocyanides in methanol at room temperature to give imidazopyrazine derivatives in excellent yields. The imidazole nitrogen atom was able to intercept the nascent nitrilium ion, channeling the reaction toward to the sole formation of imidazopyrazines, suppressing the competitive formation of other possible side products deriving from the reaction with the high-energy nitrilium ion. The number of examples and the variability of the nature of isocyanides, aldehydes, and amine components herein employed, witness the robustness of this novel methodology.

## 1. Introduction

Although multistep synthesis is the only possible choice for the preparation of highly complex natural products, it is important to keep in mind the actual feasibility and pragmatic aspects of those synthetic sequences. A recent paper [1] demonstrated that for the famous 49 steps Woodward’s synthesis of chlorophyll a, 1 mg of product would have required 230 kg of ethyl acetoacetate, the first chemical used in that project. Therefore, it is important, for concepts like “efficiency of synthesis” [2] and “reaction mass efficiency” [3], to consider the real practical application of any synthetic project. For this reason, the identification of novel reactions enabling the one-pot assembly of compounds of medium structural complexity is an important field of research with significant implications for all chemistry branches [4]. Multicomponent reactions (MCRs) are among these types of reactions, allowing for the formation of three or four covalent bonds in one single chemical operation with a net reduction of time and chemical waste. When providing the desired chemical output, MCRs usually represent the most efficient choice available to chemists [5,6,7,8,9,10,11,12,13,14,15].

Two more advantages of these MCRs are the rapid generation of a library of compounds, and the possibility to generate molecular scaffolds, which are difficult or in some cases impossible to obtain by the classical two-component chemistry [16]. Finally, when compared to a multistep synthesis, the excellence of a multicomponent reaction can be easily and rapidly ascertained by a large number of chemists shortly after publication. Characteristics such as yield, ease of purification, and versatility can be easily proved, and their scope expanded, beyond the original design [17,18].

One of the most important and versatile multicomponent reactions available to chemists is the Ugi reaction [8], a four-component one-pot reaction among equimolar amounts of amines (**1**), aldehydes (**2**), isocyanides (**3**), and carboxylic acids (**5**) which yields α-acylaminoamides (**6**) under mild reaction conditions in an alcoholic solvent (Scheme 1).

Thanks to its versatility, this reaction can also be used as the conceptual starting point for the discovery of novel MCRs, for example, by replacing one of the components with a suitable isostere [19]. Another possibility consists in removing the carboxylic acid, allowing the nitrilium ion intermediate (**4**) to be intramolecularly intercepted by a passive nucleophile unable to interfere with the initial elementary steps of the Ugi reaction. This strategy is known as interrupted Ugi reaction and, despite its intrinsic potentiality, it has not been fully exploited by the scientific community [20,21,22,23,24,25,26,27,28]. The interrupted Ugi reaction could be a very powerful tool to create sets of multi-functionalized and drug-like heterocycles, as exemplified in Figure 1.

Herein, we would like to disclose a novel interrupted Ugi reaction, exploiting for the first time the imidazole ring, as a soft nucleophile able to intramolecularly intercept the nascent nitrilium ion. This intermediate enables the formation of otherwise synthetically challenging substituted imidazopyrazines, in excellent yields and under mild reaction conditions (r.t. in MeOH) Multicomponent reactions represent a powerful chemical tool to expedite medicinal chemistry and early drug discovery programs [29], especially when combined with automation and flow synthesis [30,31,32].

## 2. Results and Discussion

To perform this task, we synthesized the corresponding imidazole *N*-substituted methanamines via reductive amination between formylimidazole derivatives (**7**–**9**) and primary amines (Scheme 2), preparing eight different building blocks (**13**–**20**) (Figure 2).

With the different imidazole *N*-substituted methanamines, we then carried out the MCR using twelve different isocyanides (**21**–**32**) and twelve aldehydes (**33**–**44**) in methanol, at room temperature overnight (Figure 3).

To our delight, all the attempted reactions consistently resulted in the formation of the desired imidazopyrazines (**45**–**80**), usually in excellent yields, without the formation of any by-products (Figure 4).

In detail, yields ranged from 95% to 25% and showed to be unaffected by the nature of isocyanides. Indeed, we were able to successfully use all kinds of isocyanides: aromatic, primary, secondary, and tertiary aliphatic, and isocyanoacetate, without noting a significant decrease in yield. Aliphatic and aromatic aldehydes containing either electron-withdrawing or electron-donating groups reacted in similar ways. A reduction in yield was observed above all with heterocyclic aldehydes **35** and **37** (Figure 3). Except for compounds **77** and **78**, which were obtained as a mixture of isomers, we have generally observed the formation of one single geometrical isomer for the amidines. To further confirm this data, an X-ray crystallographic analysis of compound **79** was also performed (Figure 5).

The crystallographic data showed that the amidine adopts an E configuration about the C6 = N3 double bond; the piperazine ring is characterized by an almost half-chair conformation (Cremer–Pople puckering parameters [34] are: QT = 0.428(1) Å, θ = 46.3(5)° and φ = −160(1)°). The chloro-benzene substituent is in axial conformation and it is perpendicularly oriented with respect to the imidazole moiety (dihedral angle of 89(1)°). The bicyclic core presents a certain degree of electron delocalization, as shown by the shortening of the N2-C6 1.434(2) Å and C3-C4 1.498(2) Å bond lengths. The conformation of N4-side chain is mainly characterized by the torsional angles ω1 C4-N4-C17-C18 of 60.8(1)°, ω2 N4-C17-C18-C19 of 53.4(1)°, and ω3 C18-C19-C20-C21 of 63.6(1)°.

In the light of these results, in Scheme 3 we propose the following reaction mechanism: the secondary amine **A** reacts with the carbonyl compound **B** in methanol, forming the iminium ion **C** after the loss of a molecule of water [35]. The iminium ion cannot undergo intramolecular cyclization through the nitrogen atom of imidazole, being disfavored by an incorrect orbital overlapping [36], thus allowing for the isocyanide **D** to react, forming the highly energetic nitrilium ion **E**. A fast and irreversible 6-*exo*-*dig* intramolecular trapping by means of the nitrogen atom of imidazole affords compound **F**. As suggested by one of the reviewers as many 5-*endo*-*trig* processes, although unfavoured, are observed, it could be possible that the putative aminal formed via a 5-*endo*-*trig* cyclization may undergo reopening and can be irreversibly trapped by the isocyanide. Note that the nitrogen atom attacks the nitrilium ion in a stereoselective way [37,38], affording the amidines in *Z* configuration. Finally, due to a peri interaction, compound **F** adopts a stable *E* configuration **G** by nitrogen inversion.

## 3. Materials and Methods

### 3.1. Solvents and Reagents

Commercially available solvents and reagents were used without further purification. Tetrahydrofuran (THF) was distilled immediately before use from Na/benzophenone under a slight positive atmosphere of dry nitrogen. When needed, the reactions were performed in oven-dried glassware under a positive pressure of dry nitrogen.

### 3.2. Chromatography

Column chromatography was performed on silica gel (Merck Kieselgel 60, 230–400 mesh ASTM) using the indicated eluents. Thin layer chromatography (TLC) was carried out on 5 × 20 cm plates with a layer thickness of 0.25 mm (Merck Silica gel 60 F254, Merck KGaA, Darmstadt, Germany). When necessary they were visualized using KMnO4 reagent or Dragendorff reagent.

### 3.3. Spectra

Infrared spectra were recorded on an FT-IR Thermo-Nicolet Avatar spectrometer with absorption maxima (νmax) recorded in wavenumbers (cm−^1^). NMR spectra were recorded using a JEOL ECP 300 MHz spectrometer. Chemical shifts (δ) are quoted in parts per million referenced to the residual solvent peak. The multiplicity of each signal is designated using the following abbreviations: s, singlet; d, doublet; t, triplet; q, quartet; quint, quintet; sext, sextet; hept, heptet; m, multiplet; br s, broad singlet. Coupling constants (J) are reported in Hertz (Hz) (for more data see Appendix A). Mass spectra were recorded on a Thermo Finningan LCQ-deca XP-plus mass spectrometer (Waltham, Massachusetts, USA) equipped with an ESI source and an ion trap detector. Melting points were determined using a Stuart Scientific SMP3 apparatus and remain uncorrected.

### 3.4. Preparation of Secondary Amines *(**13**–**20**)*

The appropriate imidazolecarboxaldehyde derivative (10 mmol, 1 equiv.) was dissolved in methanol. The amine (1 equiv.) in dry THF (5.0 mL), and activated 4 Å molecular sieves (2.00 g) were added. The reaction mixture was stirred at room temperature, under nitrogen for 12 h. Then, sodium borohydride (1.2 equiv.) was added portionwise and the reaction was stirred at room temperature. After 3 h water was added, and the mixture was stirred for 10 min. Molecular sieves were filtered off through a Celite pad. The filtrate was evaporated under reduced pressure. The residue was taken up with EtOAc and the organic layer was washed with 2N NaOH (× 1), dried over sodium sulfate and concentrated in vacuo. The crude material was purified by column chromatography using the indicated eluents.

***N*-((1*H*-imidazol-5-yl)methyl)-1-phenylmethanamine (13).** The crude material was purified by column chromatography (EtOAc/MeOH 90:10 + 1% NH_4_OH) to give the product as a white solid (1.01 g, yield 54%). ^1^H-NMR (300 MHz, CD_3_OD) *δ* 7.61 (s, 1H), 7.30–7.22 (m, 5H), 6.97 (s, 1H), 3.72 (br s, 4H); ^13^C-NMR (75 MHz, CD_3_OD) *δ* 139.3, 135.8, 135.2, 128.4, 128.3, 127.0, 117.1, 52.4, 44.4; IR (KBr) 3551, 3411, 3235, 2842, 1564, 1492, 1454, 1322, 1085, 1066 ν_max_/cm^−1^; m.p. 103–104 °C; MS (ESI) *m/z* Calcd for C_11_H_14_N_3_^+^: 188.1183; found: 188.1184 [M + H]^+^.

***N-((*1*H-*imidazol-5-yl)methyl)-1-(4-chlorophenyl)methanamine (14).** The crude material was purified by column chromatography (EtOAc/MeOH 90:10 + 1% NH_4_OH) to give the product as yellow oil (1.33 g, yield 60%). ^1^H-NMR (300 MHz, CD_3_OD) *δ* 7.63 (s, 1H), 7.33–7.28 (m, 4H), 6.99 (s, 1H), 3.71 (s, 2H), 3.70 (s, 2H); ^13^C-NMR (75 MHz, CD_3_OD) *δ* 139.5, 137.1, 136.6, 134.1, 131.3, 129.7, 118.5, 53.0, 46.0; IR (neat) 3089, 2841, 1490, 1452, 1089, 1014, 936, 802, 623 ν_max_/cm^−1^; MS (ESI) *m/z* Calcd for C_11_H_13_ClN_3_^+^: 222.0793; found: 222.0795 [M + H]^+^.

***N*-((1*H*-imidazol-5-yl)methyl)hexan-1-amine (15).** The crude material was purified by column chromatography (dichloromethane/ methanol 90:10) to give the product as a yellowish oil (1.77 g, 98% yield). ^1^H NMR (400 MHz, CD_3_OD) *δ* 7.61 (s, 1H), 6.97 (s, 1H), 3.70 (s, 2H), 2.58–2.55 (m, 2H), 1.52–1.47 (m, 2H), 1.33–1.30 (m, 6H), 0.91–0.88 (m, 3H); ^13^C NMR (100 MHz, CD_3_OD) *δ* 135.1, 134.8, 117.0, 48.2, 44.6, 31.4, 28.5, 26.6, 22.2, 13.0. IR (neat) 3088, 2925, 2855, 1466, 1089, 822, 626 ν_max_/cm^−1^; MS (ESI) *m/z* Calcd for C_10_H_20_N_3_^+^: 182.1652; found: 182.1655 [M + H]^+^.

***N*-((1*H*-imidazol-5-yl)methyl)-1-(4-methoxyphenyl)methanamine (16).** The crude material was purified by column chromatography (dichloromethane/ methanol 90:10) to give the product as a yellowish oil (0.977 g, 45% yield). ^1^H NMR (400 MHz, CD_3_OD) *δ* 7.62 (s, 1H), 7.23 (d, *J* = 8.4 Hz, 2H), 6.98 (s, 1H), 6.86 (d, *J* = 8.4 Hz, 2H), 3.76 (s, 3H), 3.70 (s, 2H), 3.67 (s, 2H); ^13^C NMR (100 MHz, CD_3_OD) *δ* 158.7, 135.7, 135.3, 131.5, 129.5, 117.5, 113.8, 55.2, 52.4, 45.0. IR (neat) 3117, 2836, 1611, 1512, 1456, 1244, 1089, 1032, 815, 626 ν_max_/cm^−1^; MS (ESI) *m/z* Calcd for C_12_H_16_N_3_O^+^: 218.1288; found: 218.1291 [M + H]^+^.

***N*-((1*H*-imidazol-5-yl)methyl)aniline (17).** The crude material was purified by column chromatography (dichloromethane/ methanol 98:2) to give the product as a yellowish oil (1.59 g, 92% yield). ^1^H NMR (400 MHz, CD_3_OD) *δ* 7.58 (s, 1H), 7.10–7.06 (m, 2H), 6.93 (s, 1H), 6.67–6.59 (m, 3H), 4.23 (s, 2H); ^13^C NMR (100 MHz, CD_3_OD) *δ* 148.5, 128.5, 117.0, 112.9, 40.4. IR (neat) 3388, 3049, 2815, 2632, 1605, 1505, 1317, 1002, 836, 750 ν_max_/cm^−1^; Mp 127–128 °C; MS (ESI) *m/z* Calcd for C_10_H_12_N_3_^+^: 174.1026; found: 174.1028 [M + H]^+^.

***N*-((1*H*-imidazol-2-yl)methyl)-1-phenylmethanamine (18).** The crude material was purified by column chromatography (EtOAc/MeOH 95:5 + 1% NH_4_OH and EtOAc/MeOH 90:10 + 1% NH_4_OH) to give the product as yellow oil (0.972 g, yield 52%). ^1^H-NMR (300 MHz, CD_3_OD) *δ* 7.26–7.18 (m, 5H), 6.97 (s, 2H), 3.78 (s, 2H), 3.66 (s, 2H); ^13^C-NMR (75 MHz, CD_3_OD) *δ* 147.9, 140.4, 129.4 (2C), 128.2, 122.7, 53.8, 46.3; IR (neat) 2919, 1452, 1098, 986, 855, 736, 697 ν_max_/cm^−1^; MS (ESI) *m/z* Calcd for C_11_H_14_N_3_^+^: 188.1183; Found: 188.1184 [M + H]^+^.

***N*-((1*H*-benzo[d]imidazol-2-yl)methyl)-1-phenylmethanamine (19).** The crude material was purified by column chromatography (Pet/EtOAc 10:90) to give the product as yellow solid (0.640 g, yield 27%). ^1^H-NMR (300 MHz, CD_3_OD) *δ* 7.54–7.51 (m, 2H), 7.36–7.19 (m, 7H), 4.01 (s, 2H), 3.81(s, 2H); ^13^C-NMR (75 MHz, CD_3_OD) *δ* 155.0, 140.6, 129.4, 128.2, 123.4, 115.6, 54.1, 47.0; IR (neat) 3555, 3414, 3028, 2838, 1618, 1437, 1269, 1024, 750 ν_max_/cm^−1^; m.p. 143–145 °C; MS (ESI) *m/z* Calcd for C_15_H_16_N_3_^+^: 238.1339; found: 238.1339 [M + H]^+^.

***N*-((1*H*-benzo[d]imidazol-2-yl)methyl)-2-phenylethanamine (20).** The crude material was purified by column chromatography (Pet/EtOAc 20:80 and EtOAc/MeOH 90:10) to give the product as yellow solid (1.00 g, yield 40%). ^1^H-NMR (300 MHz, CD_3_OD) *δ* 7.53–7.49 (m, 2H), 7.26–7.12 (m, 7H), 4.01 (s, 2H), 2.92–2.87 (m, A_2_B_2_ system, 2H), 2.83–2.78 (m, A_2_B_2_ system, 2H); ^13^C-NMR (75 MHz, CD_3_OD) *δ* 154.9, 141.0, 129.6, 129.5, 127.2, 123.5, 115.7, 51.7, 47.5, 36.9; IR (neat) 3554, 3415, 2950, 2817, 1618, 1537, 1453, 1424, 1275, 1125, 1025, 751, 704 ν_max_/cm^−1^; m.p. 141–143 °C; MS (ESI) *m/z* Calcd for C_16_H_18_N_3_^+^: 252.1496; Found: 252.1497 [M + H]^+^.

### 3.5. Synthesis of Substituted Imidazopyrazines and Benzoimidazopyrazines *(**45**–**80**)*

The secondary amine (0.5 mmol, 1 equiv.) was dissolved in methanol (2.0 mL). Aldehyde (0.5 mmol, 1 equiv.) and isocyanide (0.5 mmol, 1 equiv.) were added and the reaction was stirred at room temperature under nitrogen for 18 h. The reaction mixture was concentrated under reduced pressure and the crude material was purified by column chromatography.

**(*E*)-*N*-(7-benzyl-6-hexyl-7,8-dihydroimidazo[1,5-*a*]pyrazin-5(6*H*)-ylidene)pentan-1-amine (45).** The crude material was purified by column chromatography (Pet/EtOAc 90:10 and Pet/EtOAc 70:30) to give the product as orange oil (153 mg, yield 75%). ^1^H-NMR (300 MHz, CDCl_3_) *δ* 8.32 (s, 1H), 7.29–7.27 (m, 5H), 6.78 (s, 1H), 4.13 (d, *J =* 17.1 Hz, AB system, 1H), 3.80–3.68 (m, 3H), 3.53 (d, *J =* 13.2 Hz, AB system, 1H), 3.24–3.22 (m, 2H), 1.78–1.16 (m, 16H), 1.06–0.88 (m, 6H); ^13^C-NMR (75 MHz, CDCl_3_) *δ* 150.0, 137.9, 132.8, 128.9, 128.5, 127.7, 124.9, 124.6, 58.9, 54.8, 48.3, 40.7, 31.8, 30.9, 29.7, 28.8, 25.9, 22.7, 22.6, 14.1. IR (neat) 2953, 2927, 2857, 1671, 1461, 1347, 1093, 923, 818, 699 ν_max_/cm^−1^; MS (ESI) *m/z* Calcd for C_24_H_37_N_4_^+^: 381.3013; Found: 381.3006 [M + H]^+^.

**(*E*)-*N*-(7-benzyl-6-hexyl-7,8-dihydroimidazo[1,5-*a*]pyrazin-5(6*H*)-ylidene)-2-phenylethanamine (46).** The crude material was purified by column chromatography (Pet/EtOAc 90:10) to give the product as orange oil (160 mg, yield 72%). ^1^H-NMR (300 MHz, CDCl_3_) *δ* 8.38, (s, 1H), 7.31–7.17 (m, 10H), 6.79 (s, 1H), 4.07 (d, *J* = 16.8 HZ, AB system, 1H), 3.75–3.47 (m, 5H), 3.25 (d, *J =* 13.2 Hz, AB system, 1H), 3.04–2.84 (m, 2H), 1.68–1.48 (m, 2H), 1.28–1.14 (m, 8H), 0.89–0.83 (m, 3H); ^13^C-NMR (75 MHz, CDCl_3_) *δ* 150.7, 140.0, 138.0, 135.4, 129.2, 128.8, 128.6, 128.5, 127.6, 126.4, 125.1, 124.6, 58.7, 55.1, 50.3, 40.4, 37.8, 31.8, 28.8 (2C), 25.9, 22.7, 14.2. IR (neat) 2925, 2856, 1671, 1460, 1347, 1089, 923, 744, 698 ν_max_/cm^−1^; MS (ESI) *m/z* Calcd for C_27_H_35_N_4_^+^: 415.2857; found: 415.2853 [M + H]^+^.

**(*E*)-*N*-(7-benzyl-6-hexyl-7,8-dihydroimidazo[1,5-*a*]pyrazin-5(6*H*)-ylidene)-1-phenylmethanamine (47).** The crude material was purified by column chromatography (Pet/EtOAc 70:30) to give the product as orange oil (118 mg, yield 55%). ^1^H-NMR (300 MHz, CDCl_3_) *δ* 8.41, (s, 1H), 7.33–7.23 (m, 10H), 6.81 (s, 1H), 4.52 (s, 2H), 4.17 (d, *J* = 16.8 Hz, AB system, 1H), 3.91–3.71 (m, 3H), 3.54 (d, *J =* 12.8 Hz, AB system, 1H), 1.82–1.20 (m, 10H), 0.97–0.89 (m, 3H); ^13^C-NMR (75 MHz, CDCl_3_) *δ* 151.4, 139.5, 137.8, 133.0, 129.0, 128.6, 127.8. 127.5, 127.1, 125.2, 124.6, 59.0, 55.1, 51.8, 40.7, 31.8, 28.8 (2C), 26.0, 22.7, 14.2; IR (neat) 2926, 1671, 1460, 1346, 1213, 1086, 922, 734, 697 ν_max_/cm^−1^; MS (ESI) *m/z* Calcd for C_26_H_33_N_4_^+^: 401.2700; found: 401.2699 [M + H]^+^.

**(*E*)-*N*-(7-benzyl-6-hexyl-7,8-dihydroimidazo[1,5-*a*]pyrazin-5(6*H*)-ylidene)-2-methylpropan-2-amine (48).** The crude material was purified by column chromatography (Pet/EtOAc 95:5 and Pet/EtOAc 70:30) to give the product as brown amorphous solid (154 mg, yield 79%). ^1^H-NMR (300 MHz, CDCl_3_) *δ* 8.26 (s, 1H), 7.29–7.26 (m, 5H), 6.71 (s, 1H), 4.08 (d, *J =* 17.1 Hz, AB system, 1H), 3.93–3.89 (m, 1H), 3.70–3.65 (m, 3H), 1.84–1.21 (m, 19H), 1.10–0.90 (m, 3H); ^13^C-NMR (75 MHz, CDCl_3_) *δ* 146.2, 138.1, 133.1, 128.4 (2C), 127.4, 124.9, 124.5, 58.5 (2C), 54.5, 39.6, 31.7, 31.5, 29.7, 28.8, 26.0, 22.6, 14.0; IR (KBr) 2958, 2924, 2858, 1670, 1455, 1344, 1198, 1095, 738, 652 ν_max_/cm^−1^; MS (ESI) *m/z* Calcd for C_23_H_35_N_4_^+^: 367.2857; found: fragment ion (loss of *2-methyl-N-methylenepropan-2-amine* moiety): 284.2121 [M + H]^+^.

**(*E*)-*N*-(7-benzyl-6-isopropyl-7,8-dihydroimidazo[1,5-*a*]pyrazin-5(6*H*)-ylidene)cyclohexanamine (49).** The crude material was purified by column chromatography (Pet/EtOAc 70:30) to give the product as yellow solid (121 mg, yield 69%). ^1^H-NMR (300 MHz, CDCl_3_) *δ* 8.26 (s, 1H), 7.30–7.25 (m, 5H), 6.72 (s, 1H), 4.11 (d, *J =* 17.4 Hz, AB system, 1H), 3.71–3.66 (m, 3H), 3.49 (d, *J =* 10.7 Hz, AB system, 1H), 3.29–3.30 (m, 1H), 1.95–1.92 (m, 1H), 1.79–1.45 (m, 6H), 1.24–1.20 (m, 7H), 0.93 (d, *J =* 6.7 Hz, 3H); ^13^C-NMR (75 MHz, CDCl_3_) *δ* 146.6, 138.2, 132.9, 128.5 (2C), 127.5, 125.3, 124.5, 62.5, 59.0, 57.4, 40.2, 34.7, 33.8, 27.9, 25.6, 24.4, 24.3, 20.2, 20.0; IR (KBr) 3033, 2922, 2850, 1672, 1618, 1455, 1340, 1245, 1214, 1090, 815 ν_max_/cm^−1^; m.p. 154–155 °C; MS (ESI) *m/z* Calcd for C_22_H_31_N_4_^+^: 351.2544; found: fragment ion (loss of *N-methylenecyclohexanamine* moiety): found: 242.1653 [M + H]^+^.

**(*E*)-*N*-(7-benzyl-6-isopropyl-7,8-dihydroimidazo[1,5-*a*]pyrazin-5(6*H*)-ylidene)cyclopropanamine (50).** The crude material was purified by column chromatography (Pet/EtOAc 70:30) to give the product as yellow solid (108 mg, yield 70%). ^1^H-NMR (300 MHz, CDCl_3_) *δ* 8.20 (s, 1H), 7.30–7.25 (m, 5H), 6.78 (s, 1H), 4.21 (d, *J =* 17.1 Hz, AB system, 1H), 3.81–3.47 (m, 4H), 2.70–2.63 (m, 1H), 2.56–2.42 (m, 1H), 2.06–1.95 (m, 1H), 1.24–0.81 (m, 9H); ^13^C-NMR (75 MHz, CDCl_3_) *δ* 148.6, 138.0, 132.5, 129.1, 128.4, 127.6, 124.8, 124.6, 61.0, 59.1, 41.4, 31.7, 28.3, 20.1 (2C), 9.6, 8.9; IR (KBr) 3002, 2960, 2925, 2870, 1668, 1454, 1359, 1344, 1246, 1090 ν_max_/cm^−1^; m.p. 107–108 °C; MS (ESI) *m/z* Calcd for C_19_H_25_N_4_^+^: 309.2074; found: 309.2072 [M + H]^+^.

**(*E*)-*N*-(7-benzyl-6-isopropyl-7,8-dihydroimidazo[1,5-*a*]pyrazin-5(6*H*)-ylidene)-2-phenylethanamine (51).** The crude material was purified by column chromatography (Pet/EtOAc 90:10 and Pet/EtOAc 70:30) to give the product as brown oil (113.5 mg, yield 61%). ^1^H-NMR (300 MHz, CDCl_3_) *δ* 8.34 (s, 1H), 7.29–7.16 (m, 10H), 6.76 (s, 1H), 4.12 (d, *J =* 17.1 Hz, AB system, 1H), 3.69 (d, *J =* 17.1 Hz, AB system, 1H), 3.54–3.47 (m, 3H), 3.34–3.20 (m, 2H), 3.05–2.84 (m, 2H), 2.05–1.90 (m, 1H), 1.14–1.11 (m, 3H), 0.86–0.83 (m, 3H); ^13^C-NMR (75 MHz, CDCl_3_) *δ* 149.6, 140.0, 138.1, 132.8, 129.2, 128.6 (2C), 128.5, 127.6, 126.4, 125.2, 124.8, 61.3, 58.6, 51.5, 40.8, 37.8, 28.4, 20.1, 19.9; IR (neat) 2960, 1672, 1494, 1456, 1353, 1218, 1103, 743, 698 ν_max_/cm^−1^; MS (ESI) *m/z* Calcd for C_24_H_29_N_4_^+^: 373.2387; found: fragment ion (loss of *N-methylene-2-phenylethanamine* moiety): found: 242.1654 [M + H]^+^.

**(*E*)-methyl2-((7-benzyl-6-isopropyl-7,8-dihydroimidazo[1,5-*a*]pyrazin-5(6*H*)-ylidene)amino)-acetate (52).** The crude material was purified by column chromatography (Pet/EtOAc 70:30) to give the product as brown oil (92 mg, yield 54%). ^1^H-NMR (300 MHz, CDCl_3_) *δ* 8.33 (s, 1H), 7.28–7.25 (m, 5H), 6.77 (s, 1H), 4.21 (d, *J =* 17.4 Hz, AB system, 1H), 4.05 (br s, 2H), 3.81–3.76 (m, 5H), 3.60 (d, *J =* 13.2 Hz, AB system, 1H), 3.27 (d, *J =* 10.7 Hz, 1H), 2.02–1.93 (m, 1H) 1.13 (d, *J =* 6.4 Hz, 3H), 0.86 (d, *J =* 7.8 Hz, 3H); ^13^C-NMR (75 MHz, CDCl_3_) *δ* 170.2, 152.9, 137.7, 133.2, 128.8, 128.6, 127.8, 125.0 (2C), 61.0, 59.0, 52.3, 51.2, 41.2, 28.3, 20.0, 19.8; IR (neat) 2961, 1743, 1678, 1465, 1358, 1198, 1178, 741 ν_max_/cm^−1^; MS (ESI) *m/z* Calcd for C_19_H_25_N_4_O_2_^+^: 341.1973; found: 341.1971 [M + H]^+^.

**(*E*)-*N*-(7-benzyl-6-hexyl-7,8-dihydroimidazo[1,2-*a*]pyrazin-5(6*H*)-ylidene)pentan-1-amine (53).** The crude material was purified by column chromatography (Pet/EtOAc 80:20) to give the product as yellow oil (129 mg, yield 68%). ^1^H-NMR (300 MHz, CDCl_3_) *δ* 7.62 (s, 1H), 7.31–7.29 (m, 5H), 7.00 (s, 1H), 4.25 (d, *J =* 17.8 Hz, AB system, 1H), 3.93 (d, *J =* 17.8 Hz, AB system, 1H), 3.89–3.76 (m, 3H), 3.58 (d, *J =* 13.2 Hz, AB system, 1H), 3.23 (t, *J =* 6.7 Hz, 2H) 1.75–1.27 (m, 15H), 0.94–0.87 (m, 6H); ^13^C-NMR (75 MHz, CDCl_3_) *δ* 150.9, 143.6, 137.7, 129.0, 128.6, 128.4, 127.8, 113.5, 59.4, 54.4, 48.7, 44.5, 31.8, 31.0, 29.7, 28.8 (2C), 26.0, 22.7, 22.6, 14.2 (2C); IR (neat) 2927, 2857, 1670, 1491, 1455, 1413, 1289, 1093, 889, 743, 698 ν_max_/cm^−1^; MS (ESI) *m/z* Calcd for C_24_H_37_N_4_^+^: 381.3013; found: 381.3006 [M + H]^+^.

**(*E*)-*N*-(7-benzyl-6-isopropyl-7,8-dihydroimidazo[1,2-*a*]pyrazin-5(6*H*)-ylidene)-1-phenylmethanamine (54).** The crude material was purified by column chromatography (Pet/EtOAc 80:20) to give the product as yellow oil (147 mg, yield 77%). ^1^H-NMR (300 MHz, CDCl_3_) *δ* 7.69 (s, 1H), 7.36–7.23 (m, 10H), 7.03 (s, 1H), 4.54–4.33 (m, 3H), 3.96–3.80 (m, 2H), 3.61–3.51 (m, 2H), 2.06–1.96 (m, 1H), 1.19 (d, *J =* 6.4 Hz, 3H), 0.93 (d, *J =* 6.4 Hz, 3H); ^13^C-NMR (75 MHz, CDCl_3_) *δ* 150.8, 143.9, 139.6, 137.7, 128.9, 128.6, 128.5 (2C), 127.8, 127.5, 127.0, 114.0, 60.7, 59.6, 53.1, 45.0, 28.6, 20.2, 20.0; IR (neat) 2961, 1671, 1491, 1410, 1289, 1258, 1100, 737, 697 ν_max_/cm^−1^; MS (ESI) *m/z* Calcd for C_23_H_27_N_4_^+^: 359.2231; found: 359.2229 [M + H]^+^.

**(*E*)-*N*-(7-benzyl-6-isopropyl-7,8-dihydroimidazo[1,2-*a*]pyrazin-5(6*H*)-ylidene)cyclopropanamine (55).** The crude material was purified by column chromatography (Pet/EtOAc 70:30) to give the product as yellow oil (91 mg, yield 56%). ^1^H-NMR (300 MHz, CDCl_3_) *δ* 7.45 (br d, 1H), 7.29–7.24 (m, 5H), 6.96 (br d, 1H), 4.30 (d, *J =* 18.0 Hz, AB system, 1H), 3.92–3.78 (m, 2H), 3.61–3.47 (m, 2H), 2.65–2.61 (m, 1H), 2.02–1.86 (m, 1H), 1.19–0.78 (m, 10H); ^13^C-NMR (75 MHz, CDCl_3_) *δ* 149.2, 143.3, 137.7, 129.4, 129.1, 128.4, 128.2, 127.6, 113.6, 60.5, 59.6, 45.1, 31.9, 28.3, 20.1, 20.0, 9.6, 8.9; IR (neat) 2962, 1665, 1530, 1488, 1415, 1367, 1289, 1261, 1087, 743, 698 ν_max_/cm^−1^; MS (ESI) *m/z* Calcd for C_19_H_25_N_4_^+^: 309.2074; found: 309.2072 [M + H]^+^. 

**(*E*)-*N*-(7-benzyl-6-phenyl-7,8-dihydroimidazo[1,2-*a*]pyrazin-5(6*H*)-ylidene)cyclohexanamine (56).** The crude material was purified by column chromatography (Pet/EtOAc 90:10 and Pet/EtOAc 70:30) to give the product as yellow oil (114 mg, yield 56%). ^1^H-NMR (300 MHz, CDCl_3_) *δ* 7.78 (br d, 1H), 7.41–7.23 (m, 10H), 7.02 (br d, 1H), 5.04 (s, 1H), 3.91–3.67 (m, 4H), 3.09–2.99 (m, 1H), 1.73–1.00 (m, 10H); ^13^C-NMR (75 MHz, CDCl_3_) *δ* 146.3, 144.3, 137.5, 134.7, 129.0, 128.9, 128.7, 128.6, 128.5, 128.1, 127.9, 113.7, 58.9, 58.7, 57.2, 44.9, 34.3, 33.6, 25.5, 24.2, 24.1; IR (neat) 2928, 2853, 1668, 1536, 1492, 1450, 1412, 1289, 1090, 740, 698 ν_max_/cm^−1^; MS (ESI) *m/z* Calcd for C_25_H_29_N_4_^+^: 385.2387; found: 385.2387 [M + H]^+^.

**(*E*)-*N*-(7-benzyl-6-isopropyl-7,8-dihydroimidazo[1,2-*a*]pyrazin-5(6*H*)-ylidene)-[1,1′-biphenyl]-4-amine (57).** The crude material was purified by column chromatography (Pet/EtOAc 90:10 and Pet/EtOAc 70:30) to give the product as yellow oil (158 mg, yield 70%). ^1^H-NMR (300 MHz, CDCl_3_) *δ* 7.77 (br d, 1H), 7.77–7.29 (m, 12H), 7.11 (br d, 1H), 6.86 (br d, 2H), 4.32–4.14 (m, 1H), 4.02–3.84 (m, 3H), 3.70–3.63 (m, 1H), 2.07–1.97 (m, 1H), 1.11 (br d, 3H), 0.82 (br s 3H); ^13^C-NMR (75 MHz, CDCl_3_) *δ* 150.9, 146.0, 144.7, 140.6, 137.8, 137.0, 129.3, 128.9, 128.7, 128.6, 127.8, 127.7, 127.2, 126.8, 120.8, 114.1, 62.8, 59.7, 44.1, 28.9, 20.4 (2C); IR (neat) 2963, 1667, 1535, 1484, 1407, 1259, 1220, 908, 848, 730, 697 ν_max_/cm^−1^; MS (ESI) *m/z* Calcd for C_28_H_29_N_4_^+^: 421.2387; found: 421.2386 [M + H]^+^.

**(*E*)-*N*-(7-benzyl-6-(pyridin-3-yl)-7,8-dihydroimidazo[1,2-a]pyrazin-5(6*H*)-ylidene)-2-methylpropan-2-amine (58).** The crude material was purified by column chromatography (Pet/EtOAc 50:50 and EtOAc) to give the product as yellow oil (86 mg, yield 45%). ^1^H-NMR (300 MHz, CDCl_3_) *δ* 8.56 (br d, 2H), 7.76 (s, 1H), 7.46–7.22 (m, 7H), 7.00 (s, 1H), 5.19 (s, 1H), 3.91–3.81 (m, 3H), 3.64 (d, *J =* 18.3 Hz, AB system, 1H), 1.12 (s, 9H); ^13^C-NMR (75 MHz, CDCl_3_) *δ* 157.8, 150.3, 149.8, 143.8, 142.5, 137.2, 135.7, 131.4, 128.8, 128.7, 128.0, 123.5, 114.1, 59.5, 58.5, 55.0, 44.7, 31.3; IR (neat) 2970, 1674, 1454, 1418, 1360, 1294, 1206, 1092, 911, 729, 700 ν_max_/cm^−1^; MS (ESI) *m/z* Calcd for C_22_H_26_N_5_^+^: 360.2183; found: 360.2181 [M + H]^+^.

**(*E*)-*N*-(7-benzyl-6-(furan-2-yl)-7,8-dihydroimidazo[1,2-*a*]pyrazin-5(6*H*)-ylidene)-2-methylpropan-2-amine (59).** The crude material was purified by column chromatography (Pet/EtOAc 90:10 and Pet/EtOAc 70:30) to give the product as yellow oil (93 mg, yield 50%). ^1^H-NMR (300 MHz, CDCl_3_) *δ* 7.63 (br d, 1H), 7.45–7.26 (m, 6H), 6.98 (br d, 1H), 6.31 (br d, 1H), 5.94–5.93 (m, 1H), 5.19 (m, 1H), 3.99–3.57 (m, 4H), 1.06 (s, 9H); ^13^C-NMR (75 MHz, CDCl_3_) *δ* 148.0, 144.7, 143.1, 142.7, 137.2, 129.3, 128.7, 128.2, 127.9, 114.2, 110.6, 110.5, 58.4, 54.9, 54.5, 46.7, 30.9; IR (neat) 2970, 1684, 1542, 1493, 1418, 1364, 1208, 1087, 1014, 742, 699 ν_max_/cm^−1^; MS (ESI) *m/z* Calcd for C_21_H_25_N_4_O^+^: 349.2023; found: 349.2022 [M + H]^+^.

**(*E*)-*N*-(7-(4-chlorobenzyl)-6-phenyl-7,8-dihydroimidazo[1,5-*a*]pyrazin-5(6*H*)-ylidene)cyclohexanamine (60).** The crude material was purified by column chromatography (Pet/EtOAc 70:30) to give the product as orange oil (167 mg, yield 80%). ^1^H-NMR (300 MHz, CDCl_3_) *δ* 8.48 (s, 1H), 7.41–7.26 (m, 9H), 6.74 (s, 1H), 4.99 (s, 1H), 3.81–3.63 (m, 4H), 3.12–3.08 (m, 1H), 1.74–0.88 (m, 10H); ^13^C-NMR (75 MHz, CDCl_3_) *δ* 145.3, 136.3, 134.7, 133.6, 133.0, 130.2, 128.9 (2C), 128.6, 128.2, 125.1, 125.0, 59.8, 57.6, 56.9, 41.1, 34.3, 33.5, 25.5, 24.2, 24.1; IR (neat) 2928, 2854, 1669, 1488, 1462, 1347, 1191, 1088, 803, 733, 699, 653 ν_max_/cm^−1^; MS (ESI) *m/z* Calcd for C_25_H_28_ClN_4_^+^: 419.1997; found: 419.1998 [M + H]^+^.

**(*E*)-1-((3*r*,5*r*,7*r*)-adamantan-1-yl)-*N*-(7-(4-chlorobenzyl)-6-phenyl-7,8-dihydroimidazo[1,5-*a*]pyrazin-5(6*H*)-ylidene)methanamine (61).** The crude material was purified by column chromatography (Pet/EtOAc 90:10 and Pet/EtOAc 70:30) to give the product as yellow oil (193 mg, yield 82%). ^1^H-NMR (300 MHz, CDCl_3_) *δ* 8.49 (s, 1H), 7.36–7.26 (m, 9H), 6.71 (s, 1H), 5.18 (s, 1H), 3.81 (s, 2H), 3.59 (s, 2H), 1.96 (s, 3H), 1.70–1.50 (m, 12H); ^13^C-NMR (75 MHz, CDCl_3_) *δ* 142.4, 136.5, 135.9, 133.6, 133.3, 130.2, 128.9, 128.8, 128.6, 128.4, 125.6, 124.9, 63.0, 57.5, 56.1, 44.0, 41.0, 36.3, 29.7; IR (neat) 2906, 2850, 1675, 1491, 1459, 1340, 1190, 1088, 909, 804, 729, 698 ν_max_/cm^−1^; MS (ESI) *m/z* Calcd for C_29_H_32_ClN_4_^+^: 471.2310; found: 471.2320 [M + H]^+^.

**(*E*)-methyl2-((7-(4-chlorobenzyl)-6-hexyl-7,8-dihydroimidazo[1,5-*a*]pyrazin-5(6*H*)-ylidene)amino)acetate (62).** The crude material was purified by column chromatography (Pet/EtOAc 80:20 and Pet/EtOAc 60:40) to give the product as yellow oil (156 mg, yield 75%). ^1^H-NMR (300 MHz, CDCl_3_) *δ* 8.35 (s, 1H), 7.297.18 (m, 4H), 6.78 (s, 1H), 4.16–4.12 (m, 3H), 3.78–3.52 (m, 7H), 1.76–1.16 (m, 10H), 0.87–0.85 (m, 3H); ^13^C-NMR (75 MHz, CDCl_3_) *δ* 170.0, 154.0, 136.3, 133.6, 133.4, 130.2, 128.8, 125.4, 124.3, 58.2, 55.5, 52.3, 50.2, 40.5, 31.7, 29.0, 28.8, 26.0, 22.6, 14.1; IR (neat) 2927, 2856, 1746, 1673, 1464, 1351, 1197, 1176, 1087, 1014, 807, 652 ν_max_/cm^−1^; MS (ESI) *m/z* Calcd for C_22_H_30_ClN_4_O_2_^+^: 417.2052; found: 417.2051[M + H]^+^.

**(*E*)-*N*-(7-(4-chlorobenzyl)-6-(furan-2-yl)-7,8-dihydroimidazo[1,5-*a*]pyrazin-5(6*H*)-ylidene)-2-methylpropan-2-amine (63).** The crude material was purified by column chromatography (Pet/EtOAc 90:10 and Pet/EtOAc 70:30) to give the product as yellow oil (88 mg, yield 46%). ^1^H-NMR (300 MHz, CDCl_3_) *δ* 8.33 (s, 1H), 7.43–7.34 (m, 5H), 6.77 (br d, 1H), 6.33–6.32 (m, 1H), 6.01 (br d, 1H), 5.13 (s, 1H), 3.83–3.54 (m, 4H) 1.10 (s, 9H); ^13^C-NMR (75 MHz, CDCl_3_) *δ* 147.8, 141.7, 135.8, 133.7, 133.1, 130.4, 128.8, 126.0, 124.0, 110.7, 110.5, 57.6, 55.6, 54.6, 43.0, 30.8; IR (neat) 2969, 1685, 1490, 1461, 1346, 1201, 1089, 1014, 809, 740, 654 ν_max_/cm^−1^; MS (ESI) *m/z* Calcd for C_21_H_24_ClN_4_O^+^: 383.1634; found: 383.1633 [M + H]^+^.

**(*E*)-*****N*-(2-benzyl-3-hexyl-2,3-dihydrobenzo[4,5]imidazo[1,2-*a*]pyrazin-4(1*H*)-ylidene)-2-methylpropan-2-amine (64).** The crude material was purified by column chromatography Pet/EtOAc 95:5 and Pet/EtOAc 80:20) to give the product as yellow oil (208 mg, yield 78%). ^1^H-NMR (300 MHz, CDCl_3_) *δ* 8.50–8.47 (m, 1H), 7.72–7.69 (m, 1H), 7.33–7.26 (m, 7H), 4.40 (d, *J* = 18.3 Hz, AB system, 1H), 4.09–4.00 (m, 2H), 3.85 (br s, 2H), 1.91–1.71 (m, 4H), 1.58–1.35 (m, 15H), 0.92–0.94 (m, 3H); ^13^C-NMR (75 MHz, CDCl_3_) *δ* 149.9, 149.3, 143.2, 137.9, 132.6, 128.6 (2C), 127.7, 123.5 (2C), 118.9, 116.6, 59.3, 59.2, 55.0, 44.1, 31.8, 30.2, 28.9 (2C), 26.2, 22.7, 14.1; IR (neat) 2961, 2927, 1670, 1539, 1448, 1361, 1343, 1164, 746 ν_max_/cm^−1^; MS (ESI) *m/z* Calcd for C_27_H_37_N_4_^+^: 417.3013; found: 417.3010 [M + H]^+^.

**(*E*)-*N*-(2-benzyl-3-(4-chlorophenyl)-2,3-dihydrobenzo[4,5]****imidazo[1,2-a]pyrazin-4(1*H*)-ylidene)pentan-1-amine (65).** The crude material was purified by column chromatography (Pet/EtOAc 95:5 and Pet/EtOAc 90:10) to give the product as yellow oil (119 mg, yield 52%). ^1^H-NMR (300 MHz, CDCl_3_) *δ* 8.64–8.62 (m, 1H), 7.75–7.73 (m, 1H), 7.40–7.20 (m, 11H), 5.10 (s, 1H), 4.02–3.95 (m, 3H), 3.77 (d, *J* = 12.8 Hz, AB system, 1H), 3.32–3.11 (m, 2H), 1.71–1.67 (m, 2H), 1.41–1.30 (m, 4H), 0.92 (t, *J* = 6.9 Hz, 3H); ^13^C-NMR (75 MHz, CDCl_3_) *δ* 150.0, 149.5, 143.1, 137.2, 134.6, 132.9, 132.2, 129.6, 129.2, 129.1, 128.9, 128.1, 124.1 (2C), 119.3, 116.7, 58.9, 58.8, 49.0, 45.5, 30.9, 29.7, 22.5, 14.1; IR (neat) 2928, 2855, 1665, 1650, 1489, 1449, 1357, 1167, 1090, 746, 698 ν_max_/cm^−1^; MS (ESI) *m/z* Calcd for C_28_H_30_ClN_4_^+^: 457.2154; found: 457.2157 [M + H]^+^.

**(*E*)-*****N*-(2-benzyl-3-hexyl-2,3-dihydrobenzo[4,5]imidazo[1,2-*a*]pyrazin-4(1*H*)-ylidene)cyclohexanamine (66).** The crude material was purified by column chromatography (Pet/EtOAc 95:5 and Pet/EtOAc 90:10) to give the product as yellow oil (130 mg, yield 59%). ^1^H-NMR (300 MHz, CDCl_3_) *δ* 8.55–8.52 (m, 1H), 7.73–7.70 (m, 1H), 7.35–7.26 (m, 7H), 4.43 (d, *J* = 18.1 Hz, AB system, 1H), 4.11 (d, *J* = 18.1 Hz, AB system, 1H), 3.98 (dd, *J*_1_ = 10.7 Hz, *J*_2_ = 4.3 Hz, 1H), 3.83 (d, *J* = 13.1 Hz, AB system, 1H), 3.70 (d, *J* = 13.1 Hz, AB system, 1H), 3.38–3.30 (m, 1H), 1.93–1.29 (m, 20H), 0.89 (t, *J* = 6.4 Hz, 3H); ^13^C-NMR (75 MHz, CDCl_3_) *δ* 150.8, 149.5, 143.1, 137.7, 132.4, 128.9, 128.5, 127.8, 123.7 (2C), 119.0, 116.8, 59.5, 56.9, 55.1, 44.8, 34.9, 34.7, 31.7, 29.8, 28.7, 25.8 (2C), 24.3, 24.2, 22.6, 14.1; IR (neat) 2926, 2853, 1666, 1540, 1449, 1346, 1173, 744, 698 ν_max_/cm^−1^; MS (ESI) *m/z* Calcd for C_29_H_39_N_4_^+^: 443.3170; found: 443.3170 [M + H]^+^.

**(*E*)-****methyl2-((3-hexyl-2-phenethyl-2,3-dihydrobenzo[4,5]imidazo[1,2-*a*]pyrazin-4(1H)-ylidene)amino)acetate (67).** The crude material was purified by column chromatography Pet/EtOAc 95:5 and Pet/EtOAc 90:10) to give the product as yellow oil (145 mg, yield 65%). ^1^H-NMR (300 MHz, CDCl_3_) *δ* 8.53–8.50 (m, 1H), 7.70–7.67 (m, 1H), 7.34–7.15 (m, 7H), 4.38 (d, *J* = 18.3 Hz, AB system, 1H), 4.33 (br d, 1H), 4.15 (d, *J* = 18.0 Hz, AB system, 1H), 3.90–3.77 (m, 5H), 2.93–2.78 (m, 4H), 1.84–1.51 (m, 3H), 1.37–1.23 (m, 7H), 0.91–0.87 (m, 3H); ^13^C-NMR (75 MHz, CDCl_3_) *δ* 170.5, 156.7, 149.3, 142.9, 139.6, 132.2, 128.9, 128.6, 126.5, 124.5, 124.3, 119.1, 116.9, 57.5, 57.1, 52.4, 50.8, 44.6, 35.2, 31.8, 29.4, 29.0, 26.2, 22.7, 14.1; IR (neat) 2928, 2885, 1746, 1670, 1451, 1375, 1200, 1174, 746, 699 ν_max_/cm^−1^; MS (ESI) *m/z* Calcd for C_27_H_35_N_4_O_2_^+^: 447.2755; found: 447.2757 [M + H]^+^.

**(*E*)-*N*-(7-hexyl-6-(4-methoxyphenyl)-7,8-dihydroimidazo[1,5-*a*]pyrazin-5(6*H*)-ylidene)-2-methylpropan-2-amine (68).** The crude material was purified by column chromatography (*n*-hexane/ EtOAc 97:3) to give the product as brownish solid (162 mg, 85% yield). ^1^H-NMR (400 MHz, CDCl_3_) δ 8.34 (s, 1H), 7.08 (d, *J* = 8.4 Hz, 2H), 6.77 (d, *J* = 8.4 Hz, 2H), 6.65 (s, 1H), 5.03 (s, 1H), 3.71 (s, 3H), 3.52 (dd, *J_ab_* = 16.4 Hz, 2H), 2.62–2.48 (m, 2H), 1.56–1.49 (m, 2H), 1.35–1.26 (m, 6H), 1.20 (s, 9H), 0.86–0.83 (m, 3H); ^13^C-NMR (100 MHz, CDCl_3_) δ 159.2, 144.2, 132.9, 129.7, 127.5, 125.8, 124.5, 113.8, 62.7, 55.1, 54.5, 53.6, 40.5, 31.6, 31.3, 28.1, 26.8, 22.5, 14.0. IR (neat) 2965, 2929, 2861, 2833, 1668, 1511, 1460, 1338, 1246, 1037, 819 ν_max_/cm^−1^; Mp 78–79 °C; MS (ESI) *m/z* Calcd for C_23_H_35_N_4_O^+^: 383.2806; found: fragment ion (loss of *2-methyl-N-methylenepropan-2-amine* moiety): 300.2072 [M + H]^+^.

**(*E*)-4-(5-(*tert*-butylimino)-7-hexyl-5,6,7,8-tetrahydroimidazo[1,5-*a*]pyrazin-6-yl)-*N*,*N*-dimethylaniline (69).** The crude material was purified by column chromatography (*n*-hexane/ EtOAc 85:15) to give the product as yellowish solid (49 mg, 25% yield). ^1^H-NMR (400 MHz, CDCl_3_) δ 8.36 (s, 1H), 7.01 (d, *J* = 8.0 Hz, 2H), 6.68 (s, 1H), 6.60 (d, *J* = 8.0 Hz, 2H), 5.03 (s, 1H), 3.57 (dd, *J_ab_* = 15.2 Hz, 2H), 2.91 (s, 6H), 2.62–2.49 (m, 2H), 1.58–1.52 (m, 2H), 1.39–1.30 (m, 6H), 1.24 (s, 9H), 0.90–0.87 (m, 3H); ^13^C-NMR (100 MHz, CDCl_3_) δ 150.0, 144.7, 132.9, 129.4, 126.1, 124.4, 122.5, 112.0, 62.8, 54.4, 53.6, 40.7, 40.3, 31.7, 31.3, 29.7, 28.2, 26.8, 22.6, 14.0. IR (neat) 3115, 2952, 2926, 2857, 1669, 1529, 1466, 1349, 1206, 1192, 811 ν_max_/cm^−1^; Mp 104–105 °C; MS (ESI) *m/z* Calcd for C_24_H_38_N_5_^+^: 396.3122; found: fragment ion (loss of *2-methyl-N-methylenepropan-2-amine* moiety): 313.2388 [M + H]^+^.

**(*E*)-*N*-(7-hexyl-6-(naphthalen-2-yl)-7,8-dihydroimidazo[1,5-*a*]pyrazin-5(6*H*)-ylidene)-2-methylpropan-2-amine (70).** The crude material was purified by column chromatography (*n*-hexane/ EtOAc 90:10) to give the product as yellowish solid (32 mg, 16% yield). ^1^H-NMR (400 MHz, CDCl_3_) δ 8.48 (s, 1H), 7.83–7.72 (m, 3H), 7.54 (s, 1H), 7.50–7.44 (m, 3H), 6.70 (s, 1H), 5.27 (s, 1H), 3.58 (dd, *J_ab_* = 16.4 Hz, 2H), 2.72–2.61 (m, 2H), 1.66–1.59 (m, 2H), 1.42–1.29 (m, 6H), 1.26 (s, 9H), 0.93–0.90 (m, 3H); ^13^C-NMR (100 MHz, CDCl_3_) δ 143.9, 140.4, 133.6, 133.0, 132.9, 128.5, 128.1, 127.5, 127.2, 126.5, 126.4, 126.3, 125.7, 124.7, 63.5, 54.7, 53.9, 40.8, 31.7, 31.3, 28.3, 26.8, 22.6, 14.1. IR (neat) 3120, 3055, 2964, 2930, 2857, 1669, 1464, 1347, 1362, 1203, 795 ν_max_/cm^−1^; Mp 108–109 °C; MS (ESI) *m/z* Calcd for C_26_H_35_N_4_^+^: 403.2857; found: fragment ion (loss of *2-methyl-N-methylenepropan-2-amine* moiety): 320.2123 [M + H]^+^.

**(*E*)-*N*-(6-(4-chlorophenyl)-7-hexyl-7,8-dihydroimidazo[1,5-*a*]pyrazin-5(6*H*)-ylidene)-4-methoxyaniline (71).** The crude material was purified by column chromatography (*n*-hexane/ EtOAc 85:15) to give the product as colorless to yellowish oil (72 mg, 33% yield). ^1^H-NMR (400 MHz, CDCl_3_) δ 8.49 (s, 1H), 7.30 (d, *J* = 8.4 Hz, 2H), 7.13 (d, *J* = 8.4 Hz, 2H), 6.82 (s, 1H), 6.77–6.74 (m, 2H), 6.60–6.58 (m, 2H), 4.77 (s, 1H), 3.76 (s, 3H), 3.66 (dd, *J_ab_* = 16.8 Hz, 2H), 2.62–2.42 (m, 2H), 1.46–1.39 (m, 2H), 1.32–1.21 (m, 6H), 0.88–0.85 (m, 3H); ^13^C-NMR (100 MHz, CDCl_3_) δ 156.7, 148.6, 139.1, 134.3, 134.0, 133.1, 129.6, 129.0, 125.6, 125.4, 121.1, 114.3, 60.7, 55.4, 53.7, 41.2, 31.6, 27.7, 26.6, 22.5, 14.0. IR (neat) 2927, 2855, 1666, 1505, 1462, 1395, 1351, 1242, 1210, 1090, 833 ν_max_/cm^−1^; MS (ESI) *m/z* Calcd for C_25_H_30_ClN_4_O^+^: 437.2103; found: 437.2107 [M + H]^+^.

**(*E*)-*N*-(7-(4-methoxybenzyl)-6-(4-methoxyphenyl)-7,8-dihydroimidazo[1,5-*a*]pyrazin-5(6*H*)-ylidene)cyclohexanamine (72).** The crude material was purified by column chromatography (*n*-hexane/ EtOAc 70:30) to give the product as yellowish oil (202 mg, 91% yield). ^1^H-NMR (400 MHz, CDCl_3_) δ 8.46 (s, 1H), 7.29 (d, *J* = 8.4 Hz, 2H), 7.15 (d, *J* = 8.4 Hz, 2H), 6.89 (d, *J* = 8.4 Hz, 2H), 6.84 (d, *J* = 8.4 Hz, 2H), 6.74 (s, 1H), 4.95 (s, 1H), 3.82 (s, 3H), 3.79 (s, 3H), 3.77–3.73 (m, 1H), 3.65–3.57 (m, 3H), 3.13–3.07 (m, 1H), 1.74–1.01 (m, 10H); ^13^C-NMR (100 MHz, CDCl_3_) δ 159.5, 159.1, 145.8, 132.8, 130.0, 129.6, 129.3, 126.8, 125.4, 124.8, 114.0, 113.9, 58.7, 57.4, 56.7, 55.2 (2C), 40.8, 34.1, 33.4, 25.4, 24.1, 24.0. IR (neat) 2928, 1676, 1598, 1509, 1459, 1345, 1249, 1201, 1033, 732, 698 ν_max_/cm^−1^; MS (ESI) *m/z* Calcd for C_27_H_33_N_4_O_2_^+^: 445.2599; found: 445.2600 [M + H]^+^.

**(*E*)-*N*-(7-(4-methoxybenzyl)-6-phenethyl-7,8-dihydroimidazo[1,5-*a*]pyrazin-5(6*H*)-ylidene)cyclohexanamine (73).** The crude material was purified by column chromatography (*n*-hexane/ EtOAc 70:30) to give the product as off-white solid (216 mg, 98% yield). ^1^H-NMR (400 MHz, CDCl_3_) δ 8.31 (s, 1H), 7.27–7.10 (m, 7H), 6.84 (d, *J* = 8.4 Hz, 2H), 6.76 (s, 1H), 4.14–4.10 (m, 1H), 3.80 (s, 3H), 3.81–3.72 (m, 2H), 3.59–3.43 (m, 2H), 2.95–2.70 (m, 3H), 2.17–2.13 (m, 1H), 1.73–1.00 (m, 11H); ^13^C-NMR (100 MHz, CDCl_3_) δ 159.2, 147.8, 140.9, 130.2, 129.8, 128.5 (4C), 126.2, 125.0, 124.4, 113.8, 58.2, 56.3, 55.3, 54.3, 39.7, 34.3, 34.0, 31.6, 31.4, 25.4, 24.2. IR (neat) 3022, 2933, 2858, 1661, 1513, 1467, 1349, 1252, 1220, 1037, 815 ν_max_/cm^−1^; Mp 119–120 °C; MS (ESI) *m/z* Calcd for C_28_H_35_N_4_O^+^: 443.2806; found: 443.2805 [M + H]^+^.

**(*E*)-*N*-(7-(4-methoxybenzyl)-6-phenyl-7,8-dihydroimidazo[1,5-*a*]pyrazin-5(6*H*)-ylidene)-2,4,4-trimethylpentan-2-amine (74).** The crude material was purified by column chromatography (*n*-hexane/ EtOAc 80:20) to give the product as colorless to yellowish oil (206 mg, 93% yield). ^1^H-NMR (400 MHz, CDCl_3_) δ 8.43 (s, 1H), 7.33–7.19 (m, 7H), 6.90 (d, *J* = 8.0 Hz, 2H), 6.73 (s, 1H), 5.14 (s, 1H), 3.82 (s, 3H), 3.73 (s, 2H), 3.57 (dd, *J_ab_* = 16.4 Hz, 2H), 1.53 (dd, *J_ab_* = 14.4 Hz, 2H), 1.17 (s, 3H), 1.03 (s, 3H), 0.95 (s, 9H); ^13^C-NMR (100 MHz, CDCl_3_) δ 159.1, 142.0, 135.6, 129.8, 129.7, 128.7, 128.6, 128.2, 125.7, 124.7, 113.9, 62.3, 58.7, 58.4, 57.5, 55.3, 40.7, 31.9, 31.8, 30.7. IR (neat) 2953, 1674, 1511, 1459, 1338, 1246, 1207, 1191, 699, 654 ν_max_/cm^−1^; MS (ESI) *m/z* Calcd for C_28_H_37_N_4_O^+^: 445.2962; found: fragment ion (loss of *2,4,4-trimethyl-N-methylenepentan-2-amine* moiety): 306.1602 [M + H]^+^.

**(*E*)-*N*-(7-(4-methoxybenzyl)-6-phenethyl-7,8-dihydroimidazo[1,5-*a*]pyrazin-5(6*H*)-ylidene)naphthalen-1-amine (75).** The crude material was purified by column chromatography (*n*-hexane/ EtOAc 70:30) to give the product as yellowish solid (182 mg, 75% yield). ^1^H-NMR (400 MHz, CDCl_3_) δ 8.47 (s, 1H), 7.81 (d, *J* = 8.0 Hz, 1H), 7.72 (d, *J* = 8.0 Hz, 1H), 7.60 (d, *J* = 7.6 Hz, 1H), 7.48–7.42 (m, 2H), 7.26 (s, 1H), 7.14 (d, *J* = 8.0 Hz, 2H), 6.96–6.59 (m, 9H), 3.99 (dd, *J_ab_* = 16.8 Hz, 2H), 3.81 (s, 3H), 3.76–3.72 (m, 1H), 3.61 (dd, *J_ab_* = 13.2 Hz, 2H), 2.68–2.48 (m, 2H), 2.11–2.04 (m, 1H), 1.85–1.76 (m, 1H); ^13^C-NMR (100 MHz, CDCl_3_) δ 159.1, 151.9, 144.1, 139.8, 134.0, 133.4, 130.7, 129.9, 129.3 (2C), 128.0, 127.9, 127.7, 127.3, 126.4, 125.9, 125.7, 124.9, 124.8, 120.6, 115.6, 113.9, 58.1, 55.3, 55.0, 39.7, 31.1, 29.8. IR (neat) 3052, 2954, 2937, 2846, 1667, 1510, 1356, 1246, 1208, 1031, 862 ν_max_/cm^−1^; Mp 136–137 °C; MS (ESI) *m/z* Calcd for C_32_H_31_N_4_O^+^: 487.2493; found: 487.2498 [M + H]^+^.

**(*E*)-2-methyl-*N*-(6-phenethyl-7-phenyl-7,8-dihydroimidazo[1,5-*a*]pyrazin-5(6*H*)-ylidene)propan-2-amine (76).** The crude material was purified by column chromatography (*n*-hexane/ EtOAc 70:30) to give the product as colorless to brownish oil (173 mg, 93% yield). ^1^H-NMR (400 MHz, CDCl_3_) δ 8.18 (s, 1H), 7.31–7.15 (m, 7H), 6.93–6.87 (m, 4H), 4.72–4.69 (m, 1H), 4.58 (s, 2H), 2.97–2.70 (m, 2H), 2.31–1.97 (m, 2H), 1.15 (s, 9H); ^13^C-NMR (100 MHz, CDCl_3_) δ 150.2, 144.8, 140.2, 129.4, 128.7, 128.5, 126.4, 125.2, 123.9, 121.6, 118.9, 58.2, 54.4, 38.9, 32.1, 31.4, 31.3. IR (neat) 2967, 2924, 1678, 1598, 1494, 1458, 1344, 1201, 924, 754, 697 ν_max_/cm^−1^; MS (ESI) *m/z* Calcd for C_24_H_29_N_4_^+^: 373.2387; found: fragment ion (loss of *2-methyl-N-methylenepropan-2-amine*): 290.1654 [M + H]^+^.

**2-methyl-*N*-(6-pentyl-7-phenyl-7,8-dihydroimidazo[1,5-*a*]pyrazin-5(6*H*)-ylidene)propan-2-amine (77).** The crude material was purified by column chromatography (*n*-hexane/ EtOAc 90:10) to give the product as yellowish oil (160.5 mg, 95% yield). ^1^H-NMR (400 MHz, CDCl_3_) main isomer δ 8.16 (s, 1H), 7.23–7.19 (m, 2H), 6.91–6.86 (m, 4H), 4.84–4.80 (m, 1H), 4.48 (dd, *J*_ab_ = 16.8 Hz, 2H), 1.96–1.87 (m, 1H), 1.65–1.52 (m, 3H), 1.37–1.17 (m, 4H), 1.27 (s, 9H), 0.86–0.83 (m, 3H); ^13^C-NMR (100 MHz, CDCl_3_) main isomer δ 156.4, 145.4, 140.6, 129.4, 127.1, 123.6, 121.3, 118.5, 59.1, 55.6, 38.7, 31.4, 31.3, 28.6, 25.9, 22.5, 13.9. IR (neat) 2959, 2930, 2870, 1672, 1493, 1456, 1361, 1201, 1103, 750, 695ν_max_/cm^−1^; MS (ESI) *m/z* Calcd for C_21_H_31_N_4_^+^: 339.2544; found: fragment ion (loss of *2-methyl-N-methylenepropan-2-amine*): 256.1810 [M + H]^+^.

***N*-(6-isobutyl-7-phenyl-7,8-dihydroimidazo[1,5-*a*]pyrazin-5(6*H*)-ylidene)cyclohexanamine (78).** The crude material was purified by column chromatography (*n*-hexane/ EtOAc 90:10) to give the product as yellowish oil (157.5 mg, 90% yield). ^1^H-NMR (400 MHz, CDCl_3_) main isomer δ 8.22 (s, 1H), 7.22–7.18 (m, 2H), 6.89–6.84 (m, 4H), 4.91–4.87 (m, 1H), 4.51 (dd, *J*_AB_ = 16.8 Hz, 2H), 3.47–3.42 (m, 1H), 1.97–1.11 (m, 13H), 0.98–0.96 (m, 6H); ^13^C-NMR (100 MHz, CDCl_3_) main isomer δ 174.4, 149.9, 133.0, 129.4, 124.9, 123.9, 121.1, 117.7, 70.07, 53.6, 43.7, 39.3, 34.1, 33.1, 25.5, 24.5, 24.4, 23.6. IR (neat) 2928, 2855, 1671, 1598, 1497, 1463, 1354, 1216, 924, 754, 693 ν_max_/cm^−1^; MS (ESI) *m/z* Calcd for C_22_H_31_N_4_^+^: 351.2544; found: 351.2543 [M + H]^+^.

**(*E*)-*N*-(6-(4-chlorophenyl)-7-hexyl-7,8-dihydroimidazo[1,5-*a*]pyrazin-5(6*H*)-ylidene)-2-methylpropan-2-amine (79).** The crude material was purified by column chromatography (*n*-hexane/ EtOAc 97:3) to give the product as an off-white solid (137 mg, 71% yield). ^1^H-NMR (400 MHz, CDCl_3_) δ 8.35 (s, 1H), 7.24 (d, *J* = 8.4 Hz, 2H), 7.15 (d, *J* = 8.4 Hz, 2H), 6.67 (s, 1H), 5.05 (s, 1H), 3.53 (dd, *J_ab_* = 16.8 Hz, 2H), 2.64–2.51 (m, 2H), 1.57–1.50 (m, 2H), 1.34–1.24 (m, 6H), 1.21 (s, 9H), 0.88–0.85 (m, 3H); ^13^C NMR (100 MHz, CDCl_3_) δ 143.4, 134.5, 134.0, 133.0, 129.9, 128.7, 125.4, 124.8, 62.7, 54.6, 53.8, 40.5, 31.6, 31.3, 28.1, 26.7, 22.6, 14.0. IR (neat) 2949, 2933, 2865, 2815, 1665, 1489, 1458, 1340, 1202, 1088, 1013 ν_max_/cm^−1^; Mp 108–109 °C; MS (ESI) *m/z* Calcd for C_22_H_32_ClN_4_^+^: 387.2310; found: fragment ion (loss of *2-methyl-N-methylenepropan-2-amine*): 304.1577 [M + H]^+^.

**(*E*)-*N*-(6-(4-chlorophenyl)-7-(4-methoxybenzyl)-7,8-dihydroimidazo[1,5-*a*]pyrazin-5(6*H*)-ylidene)cyclohexanamine (80).** The crude material was purified by column chromatography (*n*-hexane/EtOAc 70:30) to give the product as amorphous solid (205 mg, 91.5% yield). ^1^H-NMR (400 MHz, CDCl_3_) δ 8.46 (s, 1H), 7.31–7.18 (m, 6H), 6.89 (dd, *J* = 8.4 Hz, 2H), 6.75 (s, 1H), 4.93 (s, 1H), 3.82 (s, 3H), 3.77–3.59 (m, 4H), 3.08–3.02 (m, 1H), 1.75–1.01 (m, 10H); ^13^C NMR (100 MHz, CDCl_3_) δ 159.2, 144.9, 134.2, 133.7, 130.0, 129.5, 129.3, 128.9, 125.1, 125.0 (2C), 114.0, 58.4, 57.6, 56.8, 55.3, 40.9 (2C), 34.1, 33.4, 25.4, 24.1, 24.0. IR (neat) 2929, 2854, 1669, 1511, 1463, 1348, 1247, 1190, 1090, 815, 731 ν_max_/cm^−1^; MS (ESI) *m/z* Calcd for C_26_H_30_ClN_4_O^+^: 449.2103; found: 449.2106 [M + H]^+^.

## 4. Conclusions

In conclusion, in this manuscript we reported the one-pot multicomponent synthesis of substituted imidazopyrazines, starting from simple and easily available building blocks. Three covalent bonds and one six-membered ring were formed with this transformation under very mild reaction conditions and without the need of metal catalysts or protecting groups. Furthermore, the good to excellent yields, the ease of performance, and the possibility to use two different regioisomeric imidazolemethanamines and a benzimidazolic congener, make this novel transformation a very powerful tool to simply generate complexity and diversity.

This work demonstrated how, with the judicious choice of components, it is still possible to discover new productive interrupted Ugi reactions, after fifty years from the first report [39]. Furthermore, the drug-like nature of these compounds, along with their novelty, and therefore lack of intellectual properties, makes them interesting probes for medicinal chemistry applications. Studies on this topic are in progress and will be reported in due course.

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
