# Peer review of "Exploiting the Nucleophilicity of the Nitrogen Atom of Imidazoles: One-Pot Three-Component Synthesis of Imidazo-Pyrazines"

_molecules, 2019, doi:10.3390/molecules24101959_

Round 1
Reviewer 1 Report
Interesting and useful work by Tron and colleagues. The chemistry is novel and the substrate range involving imidazoles is very attractive in MedChem. I think after modifications can be published in molecules.
I see one major point, which deals with the description of the process.
In my opinion, the process is an extended Groebke-Bklackburn-Bienaymé reaction and there are clear precedents with similar systems (no imidazoles, but pyridines).
I believe authors, apart of the interrupted Ugi references, should cite the work of GBB MCRs and, especially, the Hulme (J. Org. Chem. 2014,79, 5153) and Carballares (Org Lett 2005, 7, 2329) work, where this homo-reactivity is described.
The second point, deals with the mechanistic aspects. In Scheme 3 and in the related text, it seems that the imidazole NH cyclizes, however, should be (at leat formally) the other N atom, which has the sigma lone pair, the one suitable for cyclization.
On the other hand, in this section, it is said that the iminium ion can not cyclize because of the Baldwin rules, but many 5-endo-trig processes (although unfavoured, are observed). Perhaps the main reason is that the putative aminal formed in equilibrium may undergo reopening and can be ireversibly trapped by the isocyanide.
Author Response
As suggested GBB MCRs, the Hulme (J. Org. Chem. 2014,79, 5153) and Carballares (Org Lett 2005, 7, 2329) works, have been added in the references.
Scheme 3 has been modified so that the nitrogen with the sigma lone pair attacks the nitrilium ion.
Dealing with the proposed reaction mechanism, finally, as observed, a note highlighting the possibility of a 5-endo-trig cyclization and a ring-opening allowing the attack of the isocyanide has been added to reference 13.
Reviewer 2 Report
This manuscript described the efficient synthesis of imidazo-pyrazines via Ugi-type reaction by Tron. The reactions afforded the product in moderate to high yields. The products are also correctly characterized. The reviewer suggests the manuscript could be published without any revisions.

Author Response
OK
Reviewer 3 Report
This is an interesting work describing a novel interrupted Ugi reaction with imidazole rings. The final compounds are also interesting. I recommend publication of the manuscript.
Author Response
OK